# Patients' experiences of acupuncture and counselling for depression and comorbid pain: a qualitative study nested within a randomised controlled trial

Ann Hopton, Janet Eldred, Hugh MacPherson

▶ Prepublication history and additional material is available. To view please visit the journal (http://dx.doi.org/10.1136/bmjopen-2014-005144).

▶ http://dx.doi.org/10.1136/bmjopen-2014-004964

Department of Health Sciences, University of York, York, North Yorkshire, UK

**Correspondence to**
Ann Hopton;
ann.hopton@york.ac.uk

## ABSTRACT

**Introduction:** Depression and pain frequently occur together and impact on outcomes of existing treatment for depression. Additional treatment options are required. This study aimed to explore patients' experiences of depression, the processes of change within acupuncture and counselling, and the elements that contributed to longer-term change.

**Methods:** In a substudy nested within a randomised controlled trial of acupuncture or counselling compared with usual care alone for depression, semistructured interviews of 52 purposively sampled participants were conducted and analysed using thematic analysis.

**Results:** Differences were reported by participants regarding their experience of depression with comorbid pain compared with depression alone. Along with physical symptoms often related to fatigue and sleep, participants with depression and comorbid pain generally had fewer internal and external resources available to manage their depression effectively. Those who had physical symptoms and were receiving acupuncture commonly reported that these were addressed as part of the treatment. For those receiving counselling, there was less emphasis on physical symptoms and more on help with gaining an understanding of themselves and their situation. Over the course of treatment, most participants in both groups reported receiving support to cope with depression and pain independently of treatment, with a focus on relevant lifestyle and behaviour changes. The establishment of a therapeutic relationship and their active engagement as participants were identified as important components of treatment.

**Conclusions:** Participants with and without comorbid pain received acupuncture or counselling for depression, and reported specific identifiable treatment effects. The therapeutic relationship and participants' active engagement in recovery may play distinct roles in driving long-term change. Patients who present with depression and physical symptoms of care may wish to consider a short course of acupuncture to relieve symptoms prior to a referral for counselling if needed.

**Trial registration number:** ISRCTN63787732.

### Strengths and limitations of this study

- The 52 telephone interviews were obtained from a wide range of participants in socially diverse settings and provide rich data on the participants' experiences of depression and the treatment received in the Acupuncture, Counselling or Usual Care for Depression (ACUDep) trial.
- The thematic analysis was conducted using a bottom-up process to allow the themes to develop directly from the participants' own words, and we present the positive and negative experiences of each form of treatment, whether treatment was beneficial or not.
- Our findings identify mechanisms within the processes of change that are specific to acupuncture and counselling that facilitate reduction in the symptoms of depression.
- Participants may have attributed changes directly to treatment rather than concurrent, coincidental contextual changes.
- There is a possibility of recall bias; however, it is likely that the participants recalled the aspects of treatment that were most salient to them.
- Telephone interviews prevented the interviewer gathering non-verbal contextual information, although this form of interview was more acceptable to participants than a face-to-face interview.

## INTRODUCTION

Chronic pain is commonplace in half to two-thirds of participants with major depressive disorder.[1 2] The association between pain and depression becomes stronger as the severity of either increases,[3 4] and the impact of both problems on each other plays an important role in the development and maintenance of chronicity in health problems.[5] Participants with depression and comorbid pain are difficult to diagnose, feel an increased burden of disease, tend to rely heavily on healthcare services and are more difficult to treat.[6]

Identifying and managing the pain symptoms that commonly occur alongside symptoms of depression may be important in improving depression response and remission rates.[7] Previous qualitative research reports that patients with comorbid pain and depression identify the ineffectiveness of existing pain relief strategies, a lack of tailoring strategies to meet personal needs and difficulties with patient—physician interaction as barriers to the effective self-management of their symptoms.[8] Patients appreciate a healthcare approach that is individualised[9] and are open to the value of complementary healthcare.[10] Two healthcare options that offer these attributes are acupuncture and counselling.

There is a growing evidence base in support of the effectiveness of acupuncture for a range of musculoskeletal conditions[11 12]; however, despite its widespread use by participants[13] there has been limited evidence for acupuncture as an effective treatment option for depression.[14] Patients with strong preferences for psychotherapy or counselling for depression are not likely to engage in antidepressant treatment,[15] yet the evidence for counselling as a treatment for depression is limited[16] despite widespread utilisation in primary care in the UK, with around 90% of general practices providing on-site counselling services.[17] To address this evidence gap, a randomised controlled trial Acupuncture, Counselling or Usual Care for Depression (ACUDep) compared acupuncture or counselling to usual care as treatments for primary care patients with ongoing depression.[18] The results showed that acupuncture and counselling were clinically effective in reducing depression in the short to medium term.

In a quantitative substudy nested within this trial, which focused on the effect of comorbid pain on the outcome of treatment for depression,[19] it was found that approximately 50% of the ACUDep participants had comorbid pain with depression and that the participants' pain scores at baseline predicted the outcome of treatment for depression at the 3-month follow-up point. A comparison of the treatments showed that patients with depression and comorbid pain had a better outcome in the short to medium term with acupuncture.

The ACUDep trial[18] was novel in its' investigation of acupuncture or counselling compared with usual care; yet in focusing on the clinical effectiveness of the treatments offered, the published results did not set out how the process of change occurred as a result of receiving the three treatments of acupuncture, counselling and usual care. To meet this evidence gap the aim of this qualitative study nested within the ACUDep trial is to explore patients' reports of their experiences and identify from their reports how each intervention influences long-term change.

## METHODS
### Design
This research comprised a qualitative substudy nested within a three-arm randomised controlled trial conducted between November 2009 and June 2012 of acupuncture or counselling provided as an adjunct to usual care compared with usual care alone.[18] In this nested qualitative substudy, a purposive sampling frame was prepared to recruit interviewees in the same 2:2:1 proportion of the main trial, to include 50% men and 50% women in each arm and to balance whether in pain at baseline or not. The design used a constructivist approach to grounded theory[20] to ensure that the theories were constructed by the researchers as a result of their interaction with the area of the research and the participants.

### Participants
Within the ACUDep trial,[18] 755 participants aged 18 and over with a history of on-going depression and score of 20 or more on the Beck Depression Inventory (BDI-II) were recruited by general practitioners at 27 primary care practices across Yorkshire, County Durham and Northumberland. Participants were allocated remotely by the York Trials Unit, with the allocation code concealed from the recruiting researcher, to three groups in the proportions of 2:2:1 to acupuncture, counselling and usual care alone, respectively. Within the acupuncture arm, 23 acupuncturists registered with the British Acupuncture Council for a minimum of 3 years delivered up to 12 acupuncture treatments per patient, usually weekly. Treatments were tailored to individual participants' needs within a trial protocol,[21] which included treating symptoms according to traditional Chinese medicine theory and the integration of relevant life-style support into the treatment strategy based on acupuncture-specific advice if considered appropriate by the acupuncturists. Within the counselling arm, 37 counsellors registered with the British Association of Counselling and Psychotherapy provided up to 12 sessions of humanistic counselling delivered according to a trial protocol[22] which stated: 'Counsellors will use empathy and advanced listening skills to help clients express feelings, clarify thoughts, and reframe difficulties, but they will not give advice or set homework'. Deviations from the trial protocol were permissible if considered necessary and recorded in the participant's log book. Usual care comprised the treatment that patients typically receive in primary care along with over-the-counter medication. Commonly prescribed medication included antidepressants and analgesics. Participants were followed up over a period of 12 months after randomisation by postal questionnaire at 3, 6, 9 and 12 months. On receipt of their final 12-month postal questionnaire and based on a sampling frame (see online supplement 1), a sample of participants who had previously consented to be contacted for an interview were invited to engage in a telephone interview of approximately 30 min duration. Altogether 52 people consented to a one to one, audiorecorded, semi-structured telephone interview for this study.

### Interviews

A researcher (AH) interviewed all 52 participants. The researcher was unknown to the participants prior to the interview, therefore the interview opened with an introduction designed to set the participant at ease, to reveal the context for their depression and to draw out the participant's account of treatment received as part of the trial. Prompts from a prepared topic guide were used to elicit the participants' experiences of depression and treatment (see online supplement 2).

All the interviews were conducted between February and May 2012. On average the interviews lasted approximately 25 min (range 11–46 min). Three participants were ill and felt fatigued, but were eager to participate; therefore exploration of the questions for these people was limited in time to prevent over-burdening them. To encourage participants to relax, each participant was asked to introduce themselves by speaking about things they like to do or hoped to do and then how depression had entered their lives, before moving on to the research-related questions within the topic guide. Interviews were audiotaped, transcribed verbatim and checked for accuracy. All recordings were of sufficient clarity and content that no repeat interviews were necessary. Each transcription was checked to remove any names and assigned a participant identification number.

### Analytical methods

An inductive thematic analysis[23] was used to search across the dataset of 52 transcribed interviews. Using a constructivist approach to grounded theory,[20] data were analysed initially by AH by reading each transcript several times, then annotating ideas to generate a list of potential inductive codes developed from the dataset to capture and summarise the participants' experiences. Coding was performed sequentially on each transcript, initially without software, working systematically throughout the entire dataset. As codes were identified, they were recorded and organised on an Excel spread sheet; sections of text that demonstrated that code were then added and collated moving back and forth across the dataset making comparisons with previous data in an iterative process.

Coding and extractions were checked by JE to verify that the participants' experiences were reflected and summarised accurately. As coding progressed, comparisons were made between codes and phrases and those with similar context or concepts were grouped together. This process was conducted within each interview and across interviews resulting in a codebook of 33 codes, seven subthemes and two themes associated with the experience of depression (table 1) and a codebook of 35 codes and four themes associated with the experience of the treatment (table 2). Saturation for the main themes occurred within each treatment option within five to seven interviews. The coding and identification of themes were discussed and developed throughout between AH, HM and JE.

To understand the participants' individual experiences of depression and the treatments received, the codes and themes were developed into a diagram (see online supplement 3) and populated with the participants' identity numbers. This enabled the researcher (AH) to trace each participant throughout the process and to make distinctions between those with depression and comorbid pain, and those with depression alone.

Throughout this paper the themes are illustrated with quotes that capture and embody the participants' experiences embedded within the analytical narrative as suggested by Braun and Clarke.[23] Additional illustrative quotes are set out in online supplements 4–8.

## RESULTS

### Participants recruited

Of the 755 participants randomised in the ACUDep trial, 674 completed their 12-month follow-up; 518 had consented to be interviewed and of these 518, 464 (89%) were potentially eligible to join the study. Of the 61 participants invited, four declined participation and three did not respond. A total of 52 participants, comprising 24 men and 28 women with an age range of 22–89 years (mean 46 years, SD 13.8) were interviewed. At baseline, 26 of these participants had reported having moderate or extreme pain or discomfort on the EQ5D questionnaire; these people formed the pain group, the remainder formed the no-pain comparator group. As part of the ACUDep trial, 22 of the 52 had been randomised to receive acupuncture, 20 to counselling and 10 received usual-care alone. A summary is presented in the sampling frame in online supplement 1. On average, those allocated to acupuncture attended 11 sessions (range 4–12) and those allocated to counselling attended 10 sessions (range 6–12).

### Symptoms experienced with depression and contextual factors

Participants reported a range of symptoms they experienced concurrently with depression: predominantly pain, fatigue and sleep disorders (see online supplements 3, 4). Most participants with depression and comorbid pain suffered from either persistent headaches or moderate to extreme muscular–skeletal pain that predated the onset of depression and compromised their mobility, to the extent that four men had considered suicide and one man had attempted suicide. Almost half of those in pain experienced sleep disturbances or an overwhelming loss of energy to the point where much of their time was spent in bed, withdrawing from social and day-to-day activities (see online supplement 3).

> There are times when things like headaches and neck aches prevent sleep...I sort of drift 48 hours without sleeping...I have in the past been, if you like, down for a prolonged period of time...just sleep and sleep and sleep and sleep and wake up and do something for half an

**Table 1** Coding tree; a summary of the relationships between codes, subthemes and themes for the experience of depression

| Codes | Subthemes | Themes |
|---|---|---|
| Muscular pain<br>Headaches<br>Gastrointestinal pain<br>Disability<br>Search for diagnosis | Pain | Reported symptoms of depression |
| Tired all the time<br>Exhausted-worn out<br>Debilitated<br>No energy<br>Withdrawal | Fatigue | |
| Insomnia<br>Early wakening<br>Nightmares<br>Too much sleep | Sleep disorders | |
| Low self-esteem<br>What other people think<br>Guilt and self-blame<br>Black and white thinking | Negative schema | Contextual factors |
| Family life<br>Motherhood<br>Caring<br>Bereavement<br>Isolation | Home life | |
| Shift pattern<br>Work overload<br>Drive, motivation<br>High bar, aiming high | Work strain | |
| Crime<br>Bullied/threat/physical abuse<br>Self-harm<br>Childhood | Victimhood | |

hour and then go back to sleep again. I can just about function with pain killers…The only time I did try to top myself that led to even more depression because I couldn't even do that right! I'd taken a boat load of the diazepam I was on, to try and calm me down. I'd sort of stock piled some of that…washed it all down with Glenfiddich, and instead of, you know, just shuffling off quietly, all I did was end up waking up feeling absolutely dreadful in a puddle of my own vomit, and it was one of those things where, you know, it took me weeks afterwards, thinking well, I can't even kill myself properly. (p25,M,coun)

With regards to contextual factors, for the majority of the pain group, the pain they experienced had compromised their ability to work. Very few had social support, and for some, their being at home meant that they were the family member available to take on a caring role for a relative, which incurred further stress, ill health and isolation (see online supplement 5).

Because I was off and not working I was able to have the time to look after my elderly mother and aunt who were both in their 80 s. So I was their carer for 4 years and unfortunately I lost both of them and my father 4 years ago, within 6 months of each other…My IBS is certainly related to depression yeah…When I was looking after the old dears, as I call them, I was offered an operation twice and I turned it down because I didn't want to be, you know, incapacitated and not able to look after them. Once they passed on, I was then able to address my own health problems. (p26,M,coun)

In summary, for the participants with depression and comorbid pain, the symptoms experienced impacted on the most basic level of physiological needs, and reduced their ability to engage in social activity, while the contextual factors compromised their security through reduced income. Together these factors suggest that this group of people have few internal and external resources remaining to effectively manage their depression.

**Table 2** Coding tree; a summary of the relationships between codes and themes for the positive and negative experiences of treatment; factors that influence long-term change and mediating factors

| Codes | | Theme |
|---|---|---|
| Beliefs and attitudes | Pretreatment factors | Mediating factors |
| Previous experience of treatment | | |
| Participant engagement | In treatment factors | |
| Therapeutic relationship | | |
| Change of perspective | (common to all arms) | Positive experiences |
| Reduced medication | | |
| Relaxation | (acupuncture) | |
| Complaints and symptoms treated | | |
| Self-understanding | (counselling) | |
| Acceptance of situation | | |
| Empowerment | | |
| Effective medication | (usual care) | |
| Referral to National Health Service (NHS) mental health services | | |
| Adverse events | (acupuncture) | Negative experiences |
| Terminology | | |
| Time consuming | | |
| Fear of needles | | |
| Difficulty opening up | (counselling) | |
| Clichéd phrases | | |
| Opened a can of worms | | |
| Expense | | |
| Side effects of medication | | |
| Long wait after referral | (usual care) | |
| Lack of general practitioner time and continuity | | |
| Ageism | | |
| Exercise advice | Lifestyle strategies | Long-term behavioural change |
| Dietary advice | | |
| Reduce alcohol intake | | |
| Take time out | | |
| Day-to-day structure | | |
| Distraction | Cognitive strategies | |
| Thinking strategies learned | | |
| Self-talk and stoicism | | |
| Social contact | | |
| Seeking support | | |

Of the participants who were pain free at baseline, several people complained of tension headaches or gastrointestinal symptoms they experienced at times of heightened stress and anxiety. Several others identified distinct patterns of disordered sleep: either difficulty in settling to sleep or a pattern of early wakening. Similar to the pain group, a few tended to withdraw at times when they felt particularly low in mood.

I get a lot of stomach problems actually when I feel depressed…And of course, really tired as well—very, very tired when I'm feeling down…Sometimes I find it really hard to cope with people. I can possibly be a bit grumpy sometimes, or really quiet. Because I can't really face talking to anybody on certain days…I just can't bear it. (p57,F,uc)

Regarding contextual factors, the majority in the no-pain comparator group were in full or part-time employment or were relatively affluent retired professional people. For many, their experience of depression concerned feelings of low self-esteem brought about by high expectations of themselves within their working life, or hectic social schedules. Others experienced low self-esteem and threats to their security from bullying at work or as a victim from domestic violence.

We're also dance teachers, which is supposed to be a hobby but has somehow involved taking over our lives… we're teaching three nights a week, so it's a bit of a commitment for a hobby…Because when we started teaching last May time…that did give me more of an impetus to

actually make the effort, because once you've got teaching you've got to go, because you're going to let people down. And I'm always glad that I do. If I'm having low days now, I'm always glad that I've been, because I'm concentrating on stuff that's completely outside of me or completely outside of my normal life...One of the things that's come through is that I hate letting people down. I'm very hard on myself. (p45,F,coun)

I tend to keep my things bottled up...And it really gets me down. My work suffers. My home life suffers. And everything suffers if I'm really bogged down with something...following a particular incident, or a series of incidents...I chose to or had to be the brunt for a lot of the aggression and violent behaviour that this man displayed at the time. So if you like, there was a particular traumatic...there was a starting point for it. (p38,M,coun)

In general, the no-pain comparator group experienced fewer demands physiologically. Most had their basic security and social needs met; this group had larger reserves of internal and external resources available to them to cope with their depression.

## Processes of change reported by those receiving acupuncture: positive and negative experiences

The processes of change identified within the data formed three stages: primarily, developing a therapeutic relationship; second, the individual diagnosis and treatment of symptoms; and finally, engendering changes in health behaviours. Within the pain group and the no-pain comparator group, the positive experiences tended to facilitate the process of change at each stage, while negative experiences contributed to the barriers to change. Most participants welcomed the opportunity to try something different for their depression. The acupuncturists' understanding of their symptoms and explanations of how acupuncture might help their particular problems initiated the development of a therapeutic relationship. In contrast, two people described their acupuncturist as brisk, efficient and professional, yet lacking in bedside manner.

She was very positive about things...I think you have more of an intimate relationship with the person doing that rather than just a person in an office somewhere. You're physically involved. (p11,M,acu)

They could do with a course in empathy. (p22,F,acu)

Within the second stage of treatment, for most participants, the therapeutic relationship was further fostered through the acupuncturist listening to the participants' concerns, and treating the symptoms of depression depending on what was diagnosed to be of most important to the participant at that point. Within the pain group, several people experienced relief from musculoskeletal pain which tended to last for a few hours or days after the session and improvement commonly built up over several sessions. Several also reported feeling

deeply relaxed during the sessions and an uplifting sense of well-being afterwards.

I thought it was quite a strange sort of feeling, but I sort of felt better quite quickly. And then I went to the second and the third, it was...it completely lifted my mood and it made me feel more motivated...It was almost as if a weight had been lifted off my head and all of a sudden I felt like some energy had come back. (p3,M,acu)

With regards to negative experiences, one woman reported extreme tiredness after the acupuncture session, a problem that was addressed by the acupuncturist by adjusting treatment during the next session, and one man with extreme back pain attributed needling pain to his damaged nerves from a previous injury. Both of these participants concluded that the treatment went well; however, they also reported,

It was not for me. (p4,M, acu; p7,F,acu)

she'd actually hit at the root for a problem that I have with my pain, because I think at one time she put a needle in me and I kicked her. Without wanting to I involuntarily kicked her. And she'd obviously hit upon– I think there was one time when I kicked out and more than a few occasions where she'd twitched a nerve that obviously. (p4,M,acu)

In contrast, within the no-pain group two participants who were worried about the potential pain of needling prior to starting the treatment later attributed the needling sensations to the healing process. Three men thought they would have been equally relaxed if they had just rested or gone for a massage, and two other participants found the sessions too time consuming.

A little bit sceptical as to whether the treatments (acupuncture) work anyway. So it was for me like, if I go and get a sport massage, it was like the equivalent of that... (p13,M,acu)

## Factors that influenced long-term change reported by those receiving acupuncture

As treatment progressed, many participants reported that their acupuncturist began guiding them to make changes to their lifestyle in order to engender beneficial long-term outcomes. For most people with pain, fear of pain and potential injury posed a barrier to engaging in physical activity. The majority of the pain group reported being encouraged to take up gentle exercise for their overall health and they also distracted themselves during periods of low mood.

The things, they seem so small, but they are important. Things like, getting out and going for a walk and getting some fresh air. And just opening your eyes in the mornings and trying to cope with life. (p7,F,acu)

He started about exercise, you know, how that can make you feel more up… (p8,F, acu)

I read a lot and try to keep my mind off it. Really. (p9,F,acu)

One man developed his own technique based on how he felt during the acupuncture to help him manage low moods, while another relied on monthly acupuncture treatments alone to stay well; another considered further treatments but found the cost prohibitive.

I sort of developed this technique and I don't know, it was like…The way I was feeling during the acupuncture… I sort of clung on to this feeling that, or this technique of gaining that feeling, so I remember on a couple of occasions where I was out and about walking, and thinking about things that would normally would start leading me to start feeling a bit down, but it was like I'd been given this tool in my head and I just sort of—it just sort of went onto auto-pilot. It was like pulling those feelings away and just sort of throwing them away…Well, it lasted for a while but it started subsiding. (p1,M,acu)

I go for acupuncture now once a month and I find that any more than a month and I can feel myself sort of slipping and feeling really, you know, starting to get worse again. And then I go and I feel much, much better… (p3,M,acu)

The advice given to the no-pain comparator group was qualitatively different. Acupuncturists advised on dietary change, the reduction of caffeine and alcohol, and relaxation, which varied with the presenting symptoms and by gender. Those participants with the least rapport also tended to be those who were less willing to make behavioural changes.

She also helped with giving me other things that I can do. Suggesting different foods for me to eat to make me feel more energetic…I was cold all the time and that made me feel more lethargic as well, because all I wanted to do was stay in and go to bed and stay warm. So she was suggesting that I literally ate warmer foods, and she gave me a list of sort of Chinese medicine sort of foods that they had. (p20,F,acu)

In general, the process of change evolved in three stages. The therapeutic relationship and active engagement in recovery acted as mediators of the outcome throughout each stage of the process. In the short term, acupuncture often relieved physiological symptoms of depression and of comorbid pain. Longer-term improvement in depression was developed through the participants' active engagement in health promoting behaviours, supported by a positive therapeutic relationship. Several participants with comorbid pain had less physical ability to engage in lifestyle changes and tended to be the passive recipients of care. These participants often had fewer external resources in the form of finance and social contact to manage their depression and comorbid symptoms in the longer term. Additional quotes are presented in online supplement 6.

### Processes of change reported by those receiving counselling: positive and negative experiences

Based on their previous experiences of counselling the majority of participants spoke of their low expectations of counselling. However, when engaged with the counselling process within the trial, most reported being relieved to have someone to talk to in confidence. For the pain and the no-pain groups, the process of change followed a common pathway: beginning with the participants' disclosure of personal information and being listened to.

I found that process to be very valuable…I found X was very much listening and empathising, but maybe offered interpretation a bit more than the National Health person. (p37,M,coun)

For most participants this two-way active engagement appeared to nurture a therapeutic relationship between the participant and counsellor. Four male participants welcomed the opportunity to speak to a male counsellor, a choice which put them at their ease, and facilitated the process. Three others found difficulty engaging with their counsellor and attributed this problem to a personality clash. This presented an early barrier to the process of change.

A lot of it does depend on who the counsellor is…I'm saying probably same sex works better. They probably have a clearer understanding of the male mind…I found him particularly sympathetic and, you know, very constructive. I think that was…I was very pleased with the way it went… (p40,M,coun)

Every single counselling cliché that you have about, oh it's parenting issues—she kind of wheeled them all out one after the other and they were already things that I'd thought about, considered and looked at and examined to the nth degree and then thought, no that's not the problem…It felt like she was reading a script almost—like a guidebook to deal with this kind of disorder…it was almost the complete opposite of what I felt like I needed. (p28,M,coun)

A second stage of the process of change was often identified as occurring around midway within the course of treatment. The iterative process of participants' disclosure continued, with deeper exploration of their past, which helped to clarify the participants' understanding of themselves and their situation.

At the end of it, it actually for me opened up a can of worms really, and I think it did me more harm than good. A lot of my problems, especially with low self-esteem, come from the way I was brought up by my parents and my father especially. And its stuff that I'd never addressed and it brought it all out, actually. And I

actually felt worse at the end of it...I don't feel so bad about it now, because I recognise why I am like I am and some of the problems I have, where they come from... Although I say they'd opened up a can of worms, and brought some upsetting experiences back...I think it was good to do that. Because those sort of things had been bottled up for many, many years...it's actually made me address them. (p26,M,coun)

It made me realize that I just held everything in. From being a little girl, everything that had ever bothered me it was never talked about. You know, I'm quite lucky that I've never had any real abuse or anything like that. It's just that I've got memories of being a child and things were said and it hurt. And I just locked it away. And I did that for years. (p41,F,coun)

Several participants realised what factors triggered and perpetuated negative thoughts, some of which were unfounded. For one man with chronic pain, this meant going through a grieving process for the loss of his former way of life before setting in place new ways of thinking and coping.

You know, I think that was the big thing that I got from it, you know—that I could see myself more positively after having the time with him. And understand that some of the negative thoughts that were coming to my mind were not reality, if you like. To let them sail past and focus on the good things that I've done in the past. (p38,M,coun)

Everything that defined what I was has now gone. And it took an awful lot of grief, if you like, to come round to the fact that it was worth trying again. (p25,M,coun)

The use of metaphors was particularly useful for de-cluttering unnecessary thoughts about their past, regaining perspective, setting their problems into context and focusing on what was important.

the discussions were much more free than I'd kind of anticipated they might have been, was using metaphors and analogies and stuff like that, to be able to describe things and move through things. And the pictures were just coming to me in my head, like. I had one which was sort of like a circuit board and it felt like some of the wires were not quite wired up properly and they weren't working and stuff like that. And I can kind of track the metaphors throughout the whole process and it feels like it was much more of a—like it all opened up. (p38,M,coun)

### Factors that influenced long-term change reported by those receiving counselling

The final stage in the process of change was directed towards enabling the participants to maintain progress independently. Gender differences became apparent in the coping strategies adopted; the majority of women took up health and well-being strategies. Compared with the women in the pain group, the women in the no-pain comparator group were able to use a wider range of

resources to cope with, and engage in social activities more easily. One woman recalls being given cognitive behavioural homework to overcome a particular anxiety.

She gave me sort of little exercises. I found it very difficult to walk down to a friend of mine. She lives in quite a built up area...People were sitting out in their gardens and I found it very intimidating. I didn't like it. I'd become really sweaty, short of breath walking down through her estate to go and see hear...basically she just taught me to get a grip on myself really, by pointing out, you know, that everything was going to be all right...short sharp steps really. And that I'd got the coping mechanisms and I could do it. (p32,F,coun)

Many male participants appear to have continued to practise the cognitive strategies learned within the earlier sessions, and applied them to their life outside the sessions. However, male participants with depression and comorbid pain found greater difficulty sustaining these strategies and returned to their general practitioner for further help.

In summary, the majority of participants had had previous experience of counselling; however, their initial low expectations of success receded as the course of treatment progressed. A few counsellors practiced a more directive intervention than humanistic counselling, according to the need of the participant. The process of change comprised three stages, each mediated by the quality of the relationship and the participant's active engagement. Additional quotes are presented in online supplement 7.

### Processes of change reported by those receiving usual care

Participants in all three arms received usual care throughout the trial. The process of change within usual care was less evident. Differences in the appraisal of general practitioner care were apparent: three older participants with depression and comorbid pain who were allocated to usual care alone reported a lack of understanding and continuity of general practitioner and they felt abandoned without hope. One 89-year-old woman reported:

I would never go to a doctor again. I am, because I suffer a lot of pain that I needn't have done if he'd been different...if he'd have listened to me instead of just pooh-poohing it off and saying, oh no, it's not that. If he'd have really listened to what I was saying, he could have done more for me...he's ignored what I've told him. Well, in fact he's very often just ignored it all together. Pretended I hadn't said it...You see, at my age you can't really change doctors. There's not many doctors want to take somebody on that's 90 years old, do they? When I'm having a really bad day...you know, and I feel I can't turn even to the doctors, you know, then yeah, I do get depressed. (p51,F,uc)

In contrast, the majority of participants who were pain free pointed to a relationship based on trust as being a component of their steady improvement over time. For most people, a regular monthly 10 min consultation was helpful and constructive. A few felt that their general practitioner had made additional time for them when they were most in need.

> He sees me every month. I have monthly meetings with him just to have a general chat about how things are… I get on fine with him. As I say, I can talk about just about anything with him, so I suppose in a way, he's sort of, if anything he's been maybe a counsellor for me, because, you know, I can sit and talk to him about stuff… (p58,M,uc)

All participants at some time had been prescribed antidepressant medication. The majority of participants on long-term antidepressant medication raised concerns about the side effects; participants in the pain group were particularly concerned about the potential effects of mixing medication for their other medical problems with their antidepressants. A few acknowledged that they needed antidepressants to maintain long-term stability.

> Staying on medication, it has transformed my life and made everybody else's life around me better as well. And I wish I'd have done it sooner…It must be four or five years now and yes, it's been life transforming…When I started on my medication and I could realize the difference—the two people I was, almost. (p7,F,acu)

In addition to medication, during the period of the trial general practitioners referred participants to a range of secondary services: two young women received three sessions with a mental health link worker, an intervention which had enabled them to regain control of their lives; three others had been advised to try online cognitive behavioural therapy which two found to be easily accessible and effective; one man at risk of suicide was referred for urgent psychiatric help.

> I was referred rapidly to A&E and was assessed by X the psychiatrist. And I was put on to intensive home treatment. Which was invaluable. As a condition of not being sectioned… (p59,M,uc)

Referrals for younger patients had been beneficial although the waiting times were long and not always found to be acceptable, leaving most patients without adequate support in a time of crisis, and without sufficient money to pay for private care.

> I go to the doctors and I have to wait a matter of I don't know how many months before I can get, you know, onto the counselling and you know, it's just like the moment's gone, sort of thing…Unfortunately that can cost a lot of money—I'm on benefit. I can't afford it…I don't hold out much hope. (p50,F,uc)

### Factors that influenced long-term change reported by those receiving usual care: positive and negative experiences

For those who received help via a referral, the advice followed a familiar pattern: to engage in lifestyle changes, to add structure to the daily routine, and to use distraction to reduce the focus on the symptoms and negative feelings. However, without support the stoicism of 'forcing myself' was a prominent default strategy among most usual care participants.

> I force myself to do things and then I generally feel better (p57,F,uc)

> It's forcing myself. Well, it's a survival strategy… (p59,M,uc)

Overall, the continuity of always seeing the same general practitioner was reported to be important and beneficial. By contrast, some older participants with depression and comorbid pain remained caught in a seemingly hopeless cycle of seeking diagnosis and treatment for a physical complaint and without resources to seek private healthcare services. Most patients who received acupuncture or counselling were also happy with the attention from their general practitioner, but had welcomed the additional treatment provided within the trial as an adjunct to their usual care. Online supplement 8 presents a number of representative quotes.

## DISCUSSION
### Principal findings

The participants' experiences of depression were a complex interplay of internal and contextual factors. Compared with participants with depression alone, participants with depression and comorbid pain had fewer internal and external resources available to effectively manage their depression in the longer term. Acupuncture and counselling treatments were individualised interventions that operated from different perspectives. Acupuncturists appeared to work from a more physical perspective to directly relieve the symptoms of depression as they presented and then helped the patient engage in health behaviours that had a positive influence on long-term change. In contrast, counsellors helped guide the patient to identify and confront underlying causes of depression and then find their own way forward. Usual care relied primarily on pharmacological interventions. Processes of change comprising three stages were identified within acupuncture and counselling, each with specific active components. For both interventions, participants reported that the establishment of a therapeutic relationship and their active engagement helped them develop coping strategies that in turn helped them be more effective in reducing their depression in the longer term. Gender differences were apparent; the majority of women utilised a wide range of health behaviours, distraction and social contact, while

men relied predominantly on cognitive strategies to manage unhelpful negative thought processes.

## Strengths and limitations

Qualitative analysis of participants' reports of acupuncture and counselling compared with usual care provided within a randomised controlled trial is novel. This study was nested within a 12-month randomised controlled trial of the effectiveness of acupuncture or counselling for depression compared with usual care. The 52 telephone interviews were obtained from a wide range of participants in socially diverse settings. The interviews provided rich data on the participants' experiences of depression and the treatment received in the trial. The thematic analysis was conducted using a bottom-up process to allow the themes to develop directly from the participants' own words. We have presented the positive and negative experiences of each form of treatment, whether treatment was beneficial or not, and we are able to enrich the quantitative results of effectiveness of the treatments offered with the qualitative data.

These qualitative findings are concordant with, and supplement the quantitative data[19] from the ACUDep trial which showed that participants with moderate to extreme pain at baseline had worse outcomes at 3 months for depression than the no-pain comparator group in all three treatment arms. Our findings extend the findings of the trial's quantitative data in two ways: first, they offer insight into how pain and disability may erode the internal resources available for the effective management of depression. Moreover, these limitations compromise the person's security by reducing ability to generate external resources such as financial income and social contact. Second, based directly on participants' accounts, our findings identify mechanisms within the processes of change that are specific to acupuncture and counselling that facilitate reduction in the symptoms of depression.

Our study has some limitations. Participants may have attributed changes directly to treatment rather than concurrent, coincidental contextual changes. To capture the participants' experience in the longer term the interviews were conducted after the participants completed their 12-month follow-up questionnaire. We accept there is a possibility of recall bias as it has long been known that there is a significant, stable association between depression and memory impairment[24] which may have altered what was recalled and how it was recalled. This lack of recall is reflected in the brevity of some of the interviews. Although all the research questions in the topic guide were covered in the interviews, the poor recall of some experiences did not permit a lengthy exploration. However, our aim was to learn more of the experiences of depression and treatment in the longer term and it is likely that the participants recalled the aspects of treatment that were most salient to them. The lack of face-to face contact during the telephone interviews prevented the interviewer gathering non-verbal contextual information such as social cues, body language, appearance and setting to supplement the verbal answers of the interviewees. A face-to-face interview may have resulted in slightly longer interview; nevertheless, a recorded telephone interview was convenient and might be considered to be more acceptable to participants in terms of time and anonymity.

## Comparison to other studies

That depression in the presence of pain is associated with a poorer response to treatment for the depression corresponds with previous studies of depression and pain comorbidity.[2 7] Patients with depression and comorbid pain tend to exhibit a cognitive bias specific to negative aspects of health and are more likely to report less favourable outcomes of treatment.[25 26] Many of the pain group participants had a musculoskeletal problem alongside their depression, and reported of fatigue and sleep disturbances. This cluster of symptoms has also been identified in 36% of older people suffering from osteoarthritis of the hip and knee.[27]

The characterisation of depression as a cycle of pain, fatigue and withdrawal that impacts on daily functioning and social activities is consistent with evidence showing that these factors create an enduring cycle of depression.[6 28] The cyclical nature of pain problems are known to activate catastrophic worry and accentuate the symptoms of depression; coping strategies such as relaxation and distraction techniques are a good way of regulating emotions if the pain is not too intense.[5] Older people with chronic pain and depression were identified in this study as the least satisfied with their primary care service, a finding which echoes earlier findings where patients with multiple physical symptoms and depression posed a greater clinical burden[2 7] and were perceived as 'difficult' by general practitioners.[1]

With regards to the mechanisms of change, our findings identify three clear stages within acupuncture and counselling. The establishment of the therapeutic alliance in the early stages is an essential component from the outset of treatment. This extends the findings from within a pragmatic trial of acupuncture for back pain[29 30] and supports an earlier model of the process and mechanisms that contribute to ongoing change in counselling developed from the user perspective.[31] Historically, the therapeutic alliance has been regarded pejoratively as a placebo 'feel good factor' based on the grounds that most individuals seek positive feedback to reinforce their own behaviour.[32] However, where this argument focuses on 'visiting' a therapist for advice and help, it misses the point that the intervention-specific advice and positive reinforcement used in conjunction with the participants' active engagement in their rehabilitation will activate beneficial behavioural change.[8] An earlier study found that some participants had difficulty putting self-care advice into practice, even when they were intellectually committed and suggests

that practitioners may need to follow-up more carefully on the advice they have given.[33]

## Implications for practice and future research

Previous work has advocated that the management of depression and comorbid pain should involve the treatment of physical and psychological components together, and the treatments should be customised and directed to addressing comorbidities.[34] Psychiatrists and general practitioners often feel ill-equipped to adequately manage the complex presentation of symptoms associated with depression and comorbid pain. A shift in care is required from the current focus on the medical aspects of physical health to an all-encompassing approach that takes into account the biopsychosocial effects of depression and comorbid pain.[35] Future research should investigate the effectiveness of using a sequential strategy of acupuncture for early relief of symptoms, especially where there are physical symptoms, followed by counselling to address deeper psychological issues and develop cognitive coping strategies to break out of the cycle of depression. In the meantime, for those who have depression and physical symptoms, our evidence suggests that acupuncture could be a useful initial referral option.

## CONCLUSION

Differences in the way depression is experienced by people with depression and comorbid pain impact on the participants' engagement with treatment and on the response to treatment for depression. The processes of acupuncture and counselling had specific identifiable effects that were beneficial to the majority of participants. The therapeutic relationship and participants' active engagement in recovery may play distinct roles in driving long-term management of depression and comorbid pain. This study has implications for policy-makers and providers of care for primary care patients with depression and comorbid pain. Providers of care may wish to consider a short course of acupuncture to relieve symptoms of depression in patients who present with depression and comorbid pain, prior to a referral for counselling if needed.

**Acknowledgements** The authors acknowledge the contribution of the patients, counsellors, acupuncturists and general medical practitioners; the ACUDep trial management team; and Anne Burton for transcription services.

**Contributors** AH is a research fellow from a nursing background whose research focuses on the non-pharmacological management of chronic pain and depression and also conducted the interviews and analysis, interpreted the data, drafted and revised the article. JE is a qualitative researcher and research administrator whose interests are based on feminist theology and older peoples' lives and assisted with coding, advised on analysis and gave final approval for publication. HM is a practising acupuncturist and senior research fellow specialising in the effectiveness, cost-effectiveness, mechanisms and safety in the evaluation of complementary medicine and revised critically important intellectual content and gave final approval for publication.

**Funding** This article presents independent research funded by the National Institute for Health Research (NIHR) under its Programme Grants for Applied Research Programme (RP-PG-0707–10186). The views expressed are those of the author(s) and not necessarily those of the National Health Service, the NIHR or the Department of Health. The funders, and the University of York in its role as sponsor for the study, had no role in study design, data collection and analysis, decision to publish, or preparation of the manuscript.

**Competing interests** None.

**Ethics approval** The topic guide had been piloted previously and granted ethical approval by the University of York NHS Research Ethics Committee (09/H1311/75).

**Provenance and peer review** Not commissioned; externally peer reviewed.

**Data sharing statement** No additional data are available.

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
