## [Reviewer comments · BMJ Open]

Some articles will have been accepted based in part or entirely on reviews undertaken for other BMJ Group journals. These will be reproduced where possible.

ARTICLE DETAILS

TITLE (PROVISIONAL)	Patients' experiences of acupuncture and counselling for depression and comorbid pain: a qualitative study nested within a randomised controlled trial.
AUTHORS	Hopton, Ann; Eldred, Janet; MacPherson, Hugh

VERSION 1 - REVIEW

REVIEWER	Caroline Smith National Institute for Complementary Medicine, University of Western Sydney, Australia
REVIEW RETURNED	26-Apr-2014

GENERAL COMMENTS	Overall the paper is well written and contributes to a greater understanding of the role of both acupuncture and counselling may provide to the treatment of depression. I have some comments to improve the manuscript. Introduction 1. How does other qualitative research contribute to our understanding of how pain and depression. The fourth paragraph opens with a statementnot enough is known about how the experiences of pain with and without co-morbid pain....but there is no supporting literature to identify the research gap. Methods 1. Could the research design in relation to the research methodology be stated at the start of the methods section to demonstrate its relevance to study design sampling, data collection and analysis. The rationale for the Grounded Theory needs to be stated. 2. I was surprised by the short time overall for interviews and note some interviews were very short, any explanation of this on this with exploring the research questions in depth? 3. At what point was saturation achieved. Was it achieved with the 52nd interview? 4. With use of the ground theory method, were the responses to the open ended analysed as you went, and questions changed as is appropriate with this method, and how these themes were tested using deeper analysis during interviews. Results 1. Exploration of the secondary questions may have been impacted by some participants attending a smaller number of sessions could this effect the credibility of the results?
---

REVIEWER	Harm van Marwijk VU University Medical Centre, the Netherlands
REVIEW RETURNED	06-May-2014

GENERAL COMMENTS	The paper looks at how experiences of depression, both with and without comorbid pain, change as a result of receiving the treatments of acupuncture, counselling and usual care. This aim of this qualitative study, nested within a large trial, to explore these experiences, is not immediately appealing or new. Few studies outside the UK are referenced. Whether these data are of interest to a broad audience is not clear. The secondary aim to report the aspects of treatment that patients' report might have had a positive influence on long-term change is more interesting to a wide audience perhaps. The rationale for this particular substudy and why it has been done is not very clear. What relationship do these data have to the main study? Is there some sort of dialogue between the two, or a conceptual framework (mixed methods)? I would suggest the authors to rethink their central aims better and particularly to focus more on what news these data may bring the particular BMJ (Open) audience. Their level of detail seems more consistent with a qualitative journal. How the separate interventions help change, is that the core? Perhaps a figure with all the various studies and why they were performed would inform the reader. More focus on what is new and what we do not already know about acupuncture, counselling and usual care and especially about the contrast between the three, may make the paper more innovative. Many of the quotes are not really informative now. The first part about depression perception is not surprising. The data may have been relevant from their project's perspective but not for the reader. The paper seems written from the perspective of the authors rather than the interested reader. The conclusion of their abstract I found particularly uninformative. 'The therapeutic relationship and participants' active engagement in recovery may play distinct roles in driving long term change. This study has implications for policy makers and providers of care for primary care when considering referral of patients with depression and comorbid pain.' I would advise them strongly to make that more informative. Perhaps the authors can better send this paper to a social science journal?
---

VERSION 1 – AUTHOR RESPONSE

REVIEWER NAME CAROLINE SMITH

Institution and Country National Institute for Complementary Medicine, University of Western Sydney, Australia

Please state any competing interests or state 'None declared': None declared

Thank you for the opportunity to review this manuscript. Overall the paper is well written and contributes to a greater understanding of the role of both acupuncture and counselling may provide to the treatment of depression.

Thank you for this positive feedback.

I have some comments to improve the manuscript.

Introduction

1. How does other qualitative research contribute to our understanding of how (sic) pain and

depression?

The fourth paragraph opens with a statementnot enough is known about how the experiences of pain with and without co-morbid pain....but there is no supporting literature to identify the research gap.

We thank the reviewer for pointing out the limited evidence on qualitative research presented within the introduction. To address this we have included additional evidence from three qualitative studies and inserted the following quotes into the introduction:

“Previous qualitative research reports that patients with comorbid pain and depression identify the ineffectiveness of existing pain-relief strategies, a lack of tailoring strategies to meet personal needs and difficulties with patient-physician interaction as barriers to the effective self-management of their symptoms.[8] Patients appreciate a health-care approach that is individualized[9] and are open to the value of complementary health-care.[10] Two health-care options that offer these attributes are acupuncture and counselling”.

We have removed the statement in the fourth paragraph referred to by the reviewer (above) and have amended the statement to clarify our meaning. The paragraph now reads:

“The ACUDep trial[18] was novel in its investigation of acupuncture or counselling compared to usual care; yet in focusing on the clinical effectiveness of the treatments offered, the published results did not set out how the process of change occurred as a result of receiving the three treatments of acupuncture, counselling and usual care. To meet this evidence gap the aim of this qualitative study, which is nested within the ACUDep trial, is to explore patients’ reports of their experiences and identify from their reports how each intervention influences long- term change”.

Methods

1. Could the research design in relation to the research methodology be stated at the start of the methods section to demonstrate its relevance to study design sampling. The rationale for the Grounded Theory needs to be stated.

As requested by the reviewer we have amended the start of the Methods section to demonstrate the research design and methodology’s relevance to the study design sampling, data collection and analysis and to include the rationale for using the Grounded theory. The section now reads:

Design

“This research comprised a qualitative sub-study nested within a three-arm randomised controlled trial conducted between November 2009 and June 2012 of acupuncture or counselling provided as an adjunct to usual care compared to usual care alone[18]. In this nested qualitative sub-study, a purposive sampling frame was prepared to recruit interviewees in the same 2:2:1 proportion as the main trial, to include 50% males and 50% females in each arm and to balance whether in pain at baseline or not. The design used a constructivist approach to grounded theory[20] to ensure that the theories were constructed by the researchers as a result of their interaction with the area of the research and the participants.”

2. I was surprised by the short time overall for interviews and note some interviews were very short, any explanation of this on this with exploring the research questions in depth?

With regards to an explanation of the short time for the interviews: Firstly, three participants were ill and felt fatigued, but were very eager to participate to share their views, therefore exploration of the questions for these people was limited in time to prevent over-burdening. We have inserted additional text in the methods section to read:

“On average the interviews lasted approximately 25 minutes (range 11-46 minutes). Three participants were ill and felt fatigued, but were eager to participate, therefore exploration of the questions for these people was limited in time to prevent over-burdening them”.

Secondly, we have explained in the limitations section of the discussion that our aim was to capture the participants’ long-term experiences and for this reason the interviews were conducted following the collection of the twelve month follow-up questionnaire data. We also discuss the links between memory and depression impairment, which are well known, and we explain that some participants may have had trouble recalling their experiences about the details of the actual treatment after twelve

months. We also report that the participants are likely to recall the aspects of treatment that were most salient to them.

Finally; these interviews were conducted by telephone, the lack of face-to face-contact during the telephone interviews prevented the interviewer gathering non-verbal contextual information such as social cues, body language, appearance, and setting to supplement the verbal answers of the interviewees.

For further explanation we have also inserted additional text in the limitations section of the discussion to read:

“Our study has some limitations. Participants’ may have attributed changes directly to treatment rather than concurrent, coincidental contextual changes. To capture the participants’ experience in the longer-term the interviews were conducted after the participants completed their twelve month follow up questionnaire. We accept there is a possibility of recall bias as it has long been known that there is a significant, stable association between depression and memory impairment[24] which may have altered what was recalled and how it was recalled. This lack of recall is reflected in the brevity of some of the interviews. Although all the research questions in the topic guide were covered in the interviews, the poor recall of some experiences did not permit a lengthy exploration. However, our aim was to learn more of the experiences of depression and treatment in the longer-term and it is likely that the participants recalled the aspects of treatment that were most salient to them. The lack of face-to face-contact during the telephone interviews prevented the interviewer gathering non-verbal contextual information such as social cues, body language, appearance, and setting to supplement the verbal answers of the interviewees. A face to face interview may have resulted in slightly longer interview, nevertheless, a recorded telephone interview was convenient and might be considered to be more acceptable to participants in terms of time and anonymity”.

3. At what point was saturation achieved. Was it achieved with the 52nd interview?

Within each treatment option, data saturation for the main themes occurred within 5-7 interviews. The treatment options are too diverse to expect data saturation across all 52 interviews. We have inserted the following text into the 2nd paragraph of the Analytical methods section:

“Saturation for the main themes occurred within each treatment option within 5-7 interviews”.

4. With use of the ground theory method, were the responses to the open ended analysed as you went, and questions changed as is appropriate with this method, and how these themes were tested using deeper analysis during interviews.

With regards to the reviewer’s first point: The topic guide for the questions had been subjected to ethical approval by the University of York ethics committee, and therefore we were not at liberty to change the questions ad hoc. To make this clear we have added additional text to the paragraph on the interviews within the Methods section:

“The topic guide had been piloted previously and granted ethical approval by the University of York ethics committee”.

With regards to the reviewer’s second point: Although we were not at liberty to alter the main questions, the prompts used to elicit information from the participants contained flexibility to more deeply probe the themes and from there, make constant comparisons with previous data as the interviews progressed.

The interviews section already states:

“Prompts from a prepared topic guide were used to elicit the participants’ experiences of depression and treatment”.

To add detail to the analytical method section we have inserted additional text (underlined below):

“As codes were identified, they were recorded and organised on an Excel spread-sheet; sections of

text that demonstrated that code were then added and collated moving back and forth across the data set making comparisons with previous data in an iterative process”.

Results

1. Exploration of the secondary questions may have been impacted by some participants attending a smaller number of sessions could this effect the credibility of the results?

We were interested in the positive and negative experiences of the treatments provided as part of the trial. Some participants felt well by the fourth and fifth sessions of acupuncture or counselling and did not require the full 12 sessions offered. Indeed, in the UK National Health Service counselling is limited to six sessions only, so this is not surprising. Other participants attended a smaller number of sessions because of time constraints or because they disliked the treatment or did not fully engage with their therapist. Their reports of negative experiences were equally as valuable as the positive reports; we have included them within the results section. We believe this presents a true picture of the patients' experiences of treatment and thus enhances the credibility of the results.

Within the strengths section of the discussion we already have stated:

“We have presented the positive and negative experiences of each form of treatment, whether treatment was beneficial or not,”

REVIEWER NAME HARM VAN MARWIJK

Institution and Country VU University Medical Centre, the Netherlands

Please state any competing interests or state 'None declared': None declared

The paper looks at how experiences of depression, both with and without comorbid pain, change as a result of receiving the treatments of acupuncture, counselling and usual care. This aim of this qualitative study, nested within a large trial, to explore these experiences, is not immediately appealing or new. Few studies outside the UK are referenced. Whether these data are of interest to a broad audience is not clear. The secondary aim to report the aspects of treatment that patients' report might have had a positive influence on long-term change is more interesting to a wide audience perhaps.

To broaden the interest to an international audience and include studies outside the UK, we have inserted in the Introduction the following statements (underlined below) drawn from an American study and two European studies:

“Previous qualitative research reports that patients with comorbid pain and depression identify the ineffectiveness of existing pain-relief strategies, a lack of tailoring strategies to meet personal needs and difficulties with patient-physician interaction as barriers to the effective self-management of their symptoms.[8] Patients appreciate a health-care approach that is individualized[9] and are open to the value of complementary health-care.[10] Two health-care options that offer these attributes are acupuncture and counselling”.

“There is a growing evidence base in support of the effectiveness of acupuncture for a range of musculoskeletal conditions[11][12] however, despite its widespread use by participants[13] there has been limited evidence for acupuncture as an effective treatment option for depression.[14] Patients' with strong preferences for psychotherapy or counselling for depression are not likely to engage in antidepressant treatment,[15]yet the evidence for counselling as a treatment for depression is limited[16] despite widespread utilisation in primary care in the UK, with around 90% of general practices providing on site counselling services”.[17]

We have taken note of the reviewer's point and have refined the wording of the aim (at the end of the Introduction) to reflect the core message of the study to read:

“...the aim of this qualitative study, which is nested within the ACUDep trial, is to explore patients' reports of their experiences and identify how each intervention influences long- term change.”

The rationale for this particular sub-study and why it has been done is not very clear. What relationship do these data have to the main study? Is there some sort of dialogue between the two, or a conceptual framework (mixed methods)? I would suggest the authors to rethink their central aims

better and particularly to focus more on what news these data may bring the particular BMJ (Open) audience. Their level of detail seems more consistent with a qualitative journal. How the separate interventions help change, is that the core?

The reviewer is correct in surmising that we have taken a mixed methods approach to investigate acupuncture and counselling or usual care for depression and comorbid pain. In the fourth paragraph of the Introduction we have sought to clarify: The rationale for the present study; the relationship between the main ACUDep study, the previous quantitative sub-study and this qualitative sub-study; and clarify our core aims. We have amended the Introduction to now read:

"The ACUDep trial[18] was novel in their investigation of acupuncture or counselling compared to usual care, yet in focusing on the clinical effectiveness of the treatments offered, the published results did not set out how the process of change occurred as a result of receiving the three treatments of acupuncture, counselling and usual care. To meet this evidence gap, the aim of this qualitative study nested within the ACUDep trial, is to explore patients' reports of their experiences and identify from their reports how each intervention influences long- term change.

Perhaps a figure with all the various studies and why they were performed would inform the reader. More focus on what is new and what we do not already know about acupuncture, counselling and usual care and especially about the contrast between the three, may make the paper more innovative. Many of the quotes are not really informative now. The first part about depression perception is not surprising. The data may have been relevant from their project's perspective but not for the reader. The paper seems written from the perspective of the authors rather than the interested reader. The ACUDep trial contained several sub-studies. We believe it would detract from the purpose of the present study to present a figure depicting the main trial and sub-studies, particularly as several are not yet published and are in preparation. We believe that the amendment in the fourth paragraph of the introduction now reflects the relationship between the main trial and the sub-study we report here. We agree with the reviewer that more focus on what is new, and especially the contrast between the three would enhance the paper. To this effect we have added more detail to the discussion section by inserting the following statement:

"Acupuncture and counselling treatments were individualised interventions that operated from different perspectives. Acupuncturists appeared to work from a more physical perspective to directly relieve the symptoms of depression as they presented and then helped the patient engage in health behaviours that had a positive influence on long-term change. In contrast, counsellors helped guide the patient to identify and confront underlying causes of depression and then find their own way forward. Usual care relied primarily on pharmacological interventions.

We accept that our results related to patient's experiences of depression are not wholly surprising. However in the first part of the results section it is contextually useful to establish the patients' perceptions of themselves and their problems prior to receiving treatment offered as part of the trial. This establishes that the ACUDep participants have considerable diversity of their experiences of depression and comorbid pain. Having established this patients' perception of their depression prior to treatment the Results section identifies the processes of change for each treatment, which can be compared with each other and with the process of change within usual care.

The quotes within the text of the results are evidence of the qualitative research and inform the interested reader of the patients' experiences using their own words within each stage of the process of change rather than the inference of the researchers. Furthermore, the figure in Supplement 3 depicts patients' initial experiences of depression along with experiences of treatment in the trial and clearly illustrates the contrasts between the three treatments. Using this figure, the interested reader can trace the experiences and outcomes of individual patients with and without comorbid pain and the outcomes they experienced.

The conclusion of their abstract I found particularly uninformative. 'The therapeutic relationship and participants' active engagement in recovery may play distinct roles in driving long term change. This study has implications for policy makers and providers of care for primary care when considering

referral of patients with depression and comorbid pain.' I would advise them strongly to make that more informative.

Perhaps the authors can better send this paper to a social science journal?

To address the reviewer's point regarding the conclusion within the Abstract section we have amended the conclusion in the Abstract and final Conclusion section by replacing the last sentence with the following statement:

"For patients who present with both depression and physical symptoms, providers of care may wish to consider a short course of acupuncture to relieve symptoms prior to a referral for counselling if needed."

Furthermore we have added to the last section of the Discussion the following:

"In the meantime, for those who have both depression and physical symptoms, our evidence suggests that acupuncture could be a useful initial referral option."

With regards to the choice of journal, we believe our paper is of interest to readers of the BMJ Open journal and serves to complement the quantitative 'sister' paper published by the BMJ Open earlier this month:

Hopton A, MacPherson H, Keding A, Morley S. Acupuncture, counselling or usual care for depression and comorbid pain: secondary analysis of a randomised controlled trial. *BMJ Open* 2014;4:e004964 doi:10.1136/bmjopen-2014-004964